# Biomimetic Self-Healing Cementitious Construction Materials for Smart Buildings

**DOI:** 10.3390/biomimetics5040047

**Published:** 2020-10-08

**Authors:** Kwok Wei Shah, Ghasan Fahim Huseien

**Affiliations:** Department of Building, School of Design and Environment, National University of Singapore, Singapore 117566, Singapore; eng.gassan@yahoo.com

**Keywords:** concrete durability, climate change, sustainability, self-healing technology

## Abstract

Climate change is anticipated to have a major impact on concrete structures through increasing rates of deterioration as well the impacts of extreme weather events. The deterioration can affect directly or indirectly climate change in addition to the variation in the carbon dioxide concentration, temperature and relative humidity. The deterioration that occurs from the very beginning of the service not only reduces the lifespan of the concretes but also demands more cement to maintain the durability. Meanwhile, the repair process of damaged parts is highly labor intensive and expensive. Thus, the self-healing of such damages is essential for the environmental safety and energy cost saving. The design and production of the self-healing as well as sustainable concretes are intensely researched within the construction industries. Based on these factors, this article provides the materials and methods required for a comprehensive assessment of self-healing concretes. Past developments, recent trends, environmental impacts, sustainability, merits and demerits of several methods for the production of self-healing concrete are discussed and analyzed.

## 1. Introduction

Human settlements are dependent on infrastructure systems (e.g., buildings, transport, energy, water, communication) to mediate a wide range of human activities. [1]. Globally, infrastructure made of concrete (e.g., bridges, buildings, wharves, etc.) is susceptible to deterioration due to the ever-increasing carbon dioxide levels, temperature, and relative humidity [2]. Concrete is primarily made up of cement, the production of which has major implications for the climate change, especially as it appears that the demand for cement is relentless. The durability and lifespan of concrete structures under different weather conditions are influenced greatly by the high sensitivity of the cement materials to harsh environmental conditions that can cause concrete expansion and eventual cracking [3,4]. In turn, severe structural issues can arise from the concrete cracking. Thus, exceptionally durable smart concrete based on self-repair technology has been developed in response to the ever-greater specifications regarding the durability that must be satisfied by these concrete structures [5].

The impact of climate change on human life and ecosystems can be varied and manifest either in a direct or indirect manner. These include the increasing sea levels, intensification in the rate and/or severity of extreme weather phenomena, increase in the frequency of heat waves, and enhanced rate of precipitation. Thus, from the perspective of engineering, more and more research attention must be directed towards the effect of climate change on the structural features of the concretes. Simultaneously, risk-based techniques can effectively be employed to evaluate how the viable strategies of climate adaptation produce an economic benefit. For example, a model for improving the hurricane risk evaluation was developed by Bjarnadottir et al. [6] by taking into account the effect of climate change on the intensity and/or frequency of hurricanes. Meanwhile, the stochastic method suggested by Bastidas-Arteaga et al. [7] for the concrete structures was intended to facilitate the investigation of the impact of global warming on chloride ingress, although the study was limited to the incipient phase of corrosion. In a different study, Stewart and Peng [8] undertook a preliminary risk and cost–benefit assessment related to the adaptation measures intended to counteract the impact of the carbonation of reinforced concrete structures (RCs). The effect of climate change on the concrete structure’s durability in various regions has also been explored in many recent studies. For instance, the contribution of climate change to the concrete deterioration triggered by corrosion was examined by Stewart et al. [9] and Wang et al. [10]. They considered a probabilistic method for evaluating corrosion-related cracking and spalling related to the effect of climate change on regions with distinct geographical conditions. Additionally, the impact of climate change on carbonation in Toronto and Vancouver (Canada) was the focus of the study by Talukdar et al. [11], wherein the carbonation depths were found to increase by around 45% over a period of a century. On the downside, the study disregarded the ambiguities associated with climate, materials, and models and failed to estimate the extent to which the concrete infrastructure deterioration and safety were influenced by the carbonation.

The durability of a material is typically aided by the self-repair capability, especially in challenging situations for human intervention, like constructions situated in areas with deleterious physical and chemical conditions [12,13,14]. Another use of self-repair is the protection of material properties, particularly in kinetic and thermodynamic conditions supporting large defect densities, like nanostructures. Nanomaterials always have superior functional characteristics, and, unlike traditional materials, they display a quicker rate of deterioration because of the high content of interfacial atoms. A wide range of nanosystems can be created through the integration of numerous functional nanostructures with some constituents added to provide the self-repairing capability. Indeed, by comparison to the development of a nanosystem with greater robustness, such an approach is quite basic [15]. The fast innovations in nanoscience and nanotechnology have recently revolutionized the development of materials with self-repair capability, including nanomaterials, which consist of particles of less than 500 nm in size. In such materials, degeneration recovery occurs without external intervention. However, external stimuli (e.g., temperature) can activate self-repair as well, which is the case with systems known as non-autonomic self-healing materials [16].

The purpose of the present study is to review the current investigations on concrete structures based on various nanomaterials with self-repair capabilities and their implications for future uses in sustainable projects. In short, this presentation consists of three parts. First, it is argued that unlike standard concretes, self-repairing concretes are more eco-friendly and can reduce pollution levels, thereby enabling the construction industry to move towards a greater sustainability and intelligent development. The second part deals with several effective and detailed self-repair methods with comparative analysis of the advantages and disadvantages of each method. The last part involves the self-repairing systems that are proposed as viable options of concrete self-recovery mechanisms in response to corrosion, damage, and cracking.

## 2. Concrete Performance in Aggressive Environments

Economically, the serviceability of building materials is of great importance, especially in contemporary infrastructures and constituent parts. In the context of urban development, the most commonly employed concrete materials have to comply with the requirements of the standard codes of practice with regard to strength and durability [17,18,19,20]. For instance, the serviceability of the concrete material in use can be reduced by factors such as suboptimal planning, low capacity or overload, flaws in the material design and structures, incorrect building practices or substandard maintenance as well as the lack of knowledge about engineering [21,22]. The concrete structures used in construction industries require an extra improvement because they deteriorate rapidly during the lifespan. This deterioration is due to a range of both extrinsic and intrinsic processes related to the chemical, physical, thermal, and biological nature [2,23]. Furthermore, the impact of inadequate use and environmental conditions on how concrete performs has been highlighted in many studies [24,25,26,27]. Within construction industries, the prevention of the steel reinforcement exposure to hazardous chemicals such as corrosive agents depends significantly on the concrete. In general, the steel reinforcement becomes exposed to corrosion as a result of crack formation in the concrete, enabling harsh chemicals (e.g., chloride) to penetrate and reach the steel reinforcement bar. After reacting with water and oxygen these chemicals produce corrosion [28]. Figure 1 illustrates a basic representation of the occurrence of corrosion in reinforced concrete.

Cracks are detrimental not only in terms of facilitating corrosion, but also concerning the aesthetics because they make the porous structure in the concrete visible, expanding in size if no remedial action is taken. The hazardous chemicals can permeate concrete through the large cracks, leading to the chemical or physical deterioration of the concrete. However, it is not possible to completely prevent micro-cracks in the concrete because it is too expensive in terms of maintenance and repairs [28]. Consequently, an additional funding allocation is needed for maintenance work with regard to the necessary materials and skilled workers. Based on these factors, it is realized that materials with self-repair capability can make a substantial difference by repairing the cracks automatically and thus not only reduce the expenditure, but also increase the lifespan of concrete structures.

## 3. Crack Problems in Concrete

The formation of cracks in concrete is a major issue that must be addressed adequately. The drying-related shrinkage, thermal contraction, external or internal restraint to shortening, subgrade settlement, and overloading can all determine the formation of cracks. Although a complete avoidance of crack formation is nearly impossible, some methods are available for mitigating such issues. The main points at which cracks form are prior to and following the concrete hardening [29]. In order to prevent the future degradation that can reduce the use life of concrete structures, knowledge must be acquired about the causes and remedial measures to be adopted in relation to crack formation at those crucial moments.

The settlement of concretes is the key determinant of the formation of pre-hardening cracks. It starts when the water is lost in the plastic state. Factors including the lack of adequate vibration, high slumps associated with exceedingly wet concretes or insufficient covering of the embedded items (for example steel reinforcement) or at the margin of the concrete are responsible for the settlement cracking. In addition, plastic shrinking can engender the pre-hardening cracking as well [30]. Such cracks usually occur in the slabs prior to the final finishing and under various environmental conditions including strong wind, low humidity, and high daytime temperatures. These conditions promote the rapid evaporation of moisture from the surface and thus determine a greater surface shrinkage compared to the interior of the concrete [31]. The surface shrinkage is restrained by the interior concrete, leading to the occurrence of stresses higher than the tensile strength of the concrete which, in turn, leads to the formation of cracks at the surface. The length of the cracks related to plastic shrinkage is variable, but they frequently reach the mid-depth of the slab. Such cracks can be attenuated through fogging, which is generally implemented at the construction site.

The factors that can stimulate crack formation and post-hardening of concrete include drying-related shrinkage, thermal contraction, and subgrade settlement. The strategy of inserting regularly spaced construction joints is usually applied to avoid shrinkage and manage the crack location. For instance, the joints can be inserted in such a way as to determine the formation of the cracks in locations where they can be anticipated without difficulty. Furthermore, the number of cracks can be minimized through the introduction of horizontal steel reinforcement that can further hinder excessive crack expansion.

## 4. Sustainability of Self-Healing Concrete

Self-repair technology is a new innovation in the concrete industry that refers to materials having high quality and the ability to repair damage on their own without any external interference. In fact, this technology is introduced in order to satisfy the demands for reduction in expenditure related to concrete structure repair and maintenance [15]. Intense interest has been generated by this technology in the last ten years due to its potential use in building structures. Self-repair technology is also known as autonomic healing, autonomic repair and self-healing [32,33]. The main applications of this technology at the moment are the repair of cracks to restore mechanical strength and automatic crack repair to avoid extra financial expenditure and necessity of more raw materials [34]. 

To make the concrete structures more serviceable and expand their use life, crack repair is essential. A material with self-repair capabilities is referred to as an intelligent or smart material, meaning that it systematically integrates both the construct of information and physical indexes like strength and durability [35], affording material functionality at a higher level. Thus, a smart material is able to regulate itself, having a capacity for detection and response as well as controlled delivery of the response. The innate ability of the natural materials and their mechanical properties to adapt intelligently has been explored in earlier studies [36]. Conversely, human-made smart materials are yet to fully mature and their applications are only restricted to medicine, bionics, and aeronautics or astronautics.

Figure 2 shows a schematic diagram of the main mechanisms of self-healing concrete. The formation of calcium carbonate from calcium hydroxide, settlement of debris and loose cement particles in presence of water, hydration of un-hydrated cementitious particles, and further swelling of the hydrated cementitious matrix are depicted [37]. It is shown that a different self-healing reaction can occur depending on the self-healing agent used in the concrete. For instance, the use of bacteria as the self-healing agent can lead to the generation of calcium carbonate because of the chemical reactions between bacteria, oxygen, and water. The bacteria produce a calcite participate via these reactions. Other expansive self-healing agents can fill up the cracks via swelling of the materials, thus repairing the concrete automatically.

One type of energy technology considered with great potential in making building structures more sustainable and energy efficient is the low carbon emission and energy-efficient building material with self-repair capabilities. Sustainable development is geared towards the careful management of human activities to ensure the future survival of humanity on this planet whilst avoiding any disruption to the ecological equilibrium [38]. Therefore, economic security, environmental safety, and societal benefits are the pillars of sustainability that have to be upheld for safeguarding biodiversity and balancing ecosystems. Within the current industry-dominated world, dedicated efforts are being made in numerous fields (e.g., engineering, science, policy-making, architecture) to achieve the resourceful implementation of sustainability to attenuate the adverse effects of human activities on ecosystems. Regarding construction materials, sustainability implies a low environmental impact of the materials [39,40]. In this context, growing attention is being paid to self-repair technology because it can diminish material deterioration, expand use life, and eliminate maintenance expenditure [41,42]. Hence, self-repair technology can make construction materials and concrete more sustainable by lengthening its use life and decreasing the ordinary Portland cement (OPC) demand and consumption, improving energy efficiency and lowering pollution levels.

## 5. Mechanism of Self-Healing in Cementitious Materials

In the human body, skin and tissues are capable of repairing themselves through replacements of damaged areas based on nutrient uptake. Similarly, the essential products that serve as nutrients in the cement-based materials with self-repair capability may enable these materials to repair the damage or deterioration (Figure 3). Ample research has been conducted in recent times to discover new methods for achieving an effective self-repair material alongside the durability of the cement-based materials. Figure 4 illustrates an overview of such methods.

### 5.1. Expansive Agents and Mineral Admixtures

In a study by Kishi et al. [44], it was found that cementitious materials like Al_2_O_3_–Fe_2_O_3_-tri (Aft), Al_2_O_3_–Fe_2_O_3_-mono (Afm), and calcium carbonate (CaCO_3_) were created in cracked concrete and Ca(OH)_2_ crystal air voids. The working assumption argued for the leaching out of such hydration yields and renewed crystallization in flow water via fractures. In line with this, a range of repair agents, including expansive agents, geo-materials, and chemical mixtures together with their blends were used to assess the performance of concrete in terms of its ability to repair itself [44,45,46]. In addition, comparative analysis was undertaken between the reference specimen and the specimen with 10% cement content replaced with expansive materials consisting of C_4_A_3_S, CaSO_4_, and CaO. Results showed that the presence of expansive agents in the concrete beams nearly enabled the repair of an early 0.22 mm crack after more than 30 days with the detection of rehydration yield between the cracks. In contrast, for the standard concrete structures, partial repair of the cracks occurred during an identical interval of time. Hence, in comparison to the standard concrete, a higher efficiency was demonstrated by the expansive agent re-crystallization in the air voids for self-repair [44]. Qureshi et al. [26] provided evidence that concrete mixtures containing expansive minerals had a greater capability of repairing themselves. Representing a marker of the state of the cement mixture at a given age, the hydration degree enabled quantitative estimation of the cement mixture’s performance in terms of self-repair.

It was demonstrated that the geo-polymer was formed due to the addition of the geo-material with a content of 71.3% SiO_2_ and 15.4% Al_2_O_3_ to the expansive material through a separate polymerization of the aluminate and silicate complexes [45]. The presence of alkali metals caused the dissolution of the polymerized aluminate and silicate complexes at the alkaline pH. As uncovered by an extensive examination, the geo-polymer gel particles were less than 2 µm in size and a large number of hydro-garnet or Aft phases were produced by cracked interfacial phases associated with the original ruptured zone (Figure 5). In comparison to the hydro-garnet phase, the dense phase contained most of the altered geo-polymer gel as revealed by the Energy Dispersive X-Ray Analyzer (EDX) spectra. According to additional analysis of the chemical additives, an improvement was achieved by supplementing regular concrete enclosing the NaHCO_3_, Na_2_CO_3_, and Li_2_CO_3_. Such composition triggered the cementitious re-crystallization and concrete particle precipitation [45]. The conclusion reached was that crack self-repair could be greatly enhanced through the addition of sufficient quantities of carbonates and expansive agents.

### 5.2. Hollow Fibers

Self-repair is made possible by the hollow fibers as illustrated in Figure 6. These fibers use voids constituting a composite network matrix for the storage of certain functional constituents of the materials serving as repair agents [47,48]. When extrinsic stresses or stimuli cause concrete structure deterioration and crack formation, the functional constituents are released from the voids in which they are stored to automatically heal damage. Thus, these hollow fibers work akin to arteries in a living organism, making it possible for the materials in which they are embedded to repair themselves [48,49]. For example, the self-repair of polymeric composites is facilitated by bulk polymers [50,51,52,53]. The technique of damage visual enhancement was devised by Pang and Bond [53] for the rapid and straightforward detection of the internal damage in the composite structures. Evidence has been provided that the progress of the healing process can be monitored via the fibers containing engineering healing materials and labelled with the fluorescent dye.

The technology of crack repair in cementitious materials is based on the same mechanism as the self-healing biological functions similar to blood coagulation [54,55]. It involves the incorporation of the functional constituents in delicate fibrous vessels permeating the structural network of the concrete. The occurrence of deterioration triggers the rupture of the fibers and release of the functional repair materials that activate self-healing. Empirical work has succeeded in making concrete less porous by incorporating liquid methyl methacrylate (MMA), methacrylic acid methyl ester and reactive resin into the hollow polypropylene fibers that were subsequently inserted into the concrete [55]. Other researchers investigated the discharge of crack-bridging cementitious glue from hollow glass pipettes within the concrete instantly after the flexural test. Compared to the concrete structures without such glue constituents, those with the glue were capable to carry 20% heavier loads.

The buoyant process associated with self-repair has been discussed in various studies [54,56,57]. This process was elucidated by inserting hollow fibers in the cementitious network matrix with one end linked to the self-repair mediator and the other end tied up. The concrete mixtures were produced and inserted into glass tubes with 2 mm external diameter and 0.8 mm internal diameter [56]. The self-repair constituent included diluted (27%) and non-diluted alkali-silicate mixtures and two components integrating epoxy resin of low viscosity. Next, loading was performed until the crack mouth opening displacement (CMOD) reached 0.03–2 mm following the load removal. To determine whether the self-repair capability was enhanced, the cracked specimens were subjected to renewed curing. A strength recovery mean ratio of 1.1 and 1.5 was respectively displayed by the specimens with the repair constituent of dilute and non-dilute alkaline silica solution, unlike the specimens without repair constituent. In contrast, the strength recovery ratio did not show significant gains for the specimens with epoxy resin. In fact, it was around three times lower than the ratio associated with direct mixing and manual injection of resin into areas with cracks. It was argued that the two components had not been properly blended and stirred, that caused the resin to harden inadequately and resulting in a low repair ratio. Another possible explanation was that the pipes still contained the residual epoxy as one end was blocked.

According to Joseph [57], apart from a few negligible differences the testing protocol was identical. The conduit of repair mediator (ethyl cyanoacrylate) was represented by bent plastic tubes with 4 mm external diameter and 3 mm internal diameter. The conclusion reached was that self-repair can effectively be accomplished based on the external provision of such a repair constituent. A considerable improvement was noted in the post-crack stiffness, peak load, and ductility following the damage repair. As suggested by the observations made during and after assessment, the capillary suction and gravitational effects made ethyl cyanoacrylate a suitable adhesive agent capable of permeating an extensive area of the surfaces with cracks.

### 5.3. Bacteria as Self-Healing Agent

Biological mechanisms for repair involving the introduction of bacteria into the concrete have been recently suggested [59,60,61,62,63]. During the middle of the 1990s, a sustainable method for healing cracks in concrete was proposed by Gollapudi [64] that involved the introduction of ureolytic bacteria to speed up CaCO_3_ precipitation in the concrete micro-crack zones. Several parameters were employed to describe this process, including the amount of dissolved inorganic carbon, the pH of the material, the levels of calcium ions, and nucleation site accessibility. The walls of the bacterial cells represented the nucleation sites, while the other parameters were regulated by the bacterial metabolism [60]. Tittelboom and coworkers [60] used bacteria in the concrete to generate an enzyme called urease that could catalyze urea (CO(NH_2_)_2_) into the ammonium ions (NH_4_^+^) and carbonate radicals (CO_3_^−2^). In the chemical reactions 1 mol of urea underwent intracellular hydrolyses to 1 mol of carbonate and 1 mol of ammonia following Path I. Then, carbamate was hydrolyzed spontaneously to form one extra mole of ammonia and carbonic acid via path II. These products later formed 1 mol of bi-carbonate (HCO_3_^−^) and 2 moles of ammonium (NH_4_^+^) and hydroxide (OH^−1^) ions (Path III and IV). Path IV and V was responsible for the enhancement of pH, drifting the bicarbonate equilibrium to form carbonate ions.
CO(NH_2_)_2_ + H_2_O → NH_2_COOH+NH_3_ Path I
NH_2_COOH + H_2_O → NH_3_ + H_2_CO_3_ Path II
H_2_CO_3_ + H_2_O → HCO_3_^−^ + H^+1^ Path III
2NH_3_ + 2H_2_O → 2NH_4_^+^ + 2OH^−1^ Path IV
HCO_3_^−^ + H^+1^+ 2NH_4_^+^ + 2OH^−1^ → CO_3_^−2^ + 2NH_4_^+^ + 2H_2_O Path V

The walls of the bacterial cells had a negative charge, so cations from the surrounding environment can be accepted by the bacteria, with the deposition of Ca^+2^ on the cell wall surface. The reaction between Ca^2+^ and CO_3_^−2^ resulted in the precipitation of CaCO_3_ on the surface of the cell wall (Path VI and VII), thus providing active nucleation sites. The repair of surface cracks was made possible through this method of bacterial-based localized CaCO_3_ precipitation as illustrated in Figure 7.
Ca^+2^ + Cell → Cell-Ca^+2^ Path VI
Cell-Ca^+2^ + CO_3_^−2^ → Cell-CaCO_3_ Path VII

### 5.4. Microencapsulation

Inspired by the natural phenomena, several encapsulation materials have been made with diverse sizes, from macro- to nano-scale. At the macroscopic level, natural encapsulation in its most basic form is embodied by the bird eggs or seeds. Whereas, at the micro level natural encapsulation is embodied by an egg or seed cell [65,66,67]. The starting point of the microencapsulation development was the creation of dye-containing capsules. Numerous novel technologies have been introduced in different fields of applications [68]. Microencapsulation is not a separate component, but can be understood as insertion of solid granules of the order of micrometers, liquid drops or gases into the inert shell, affording protection against the activity of external agents [69,70]. This was the basis of microencapsulation with self-repair capability [69,71]. The mechanism of self-repair is illustrated in Figure 8a,b. The embedded microcapsules break when the cracks are formed, causing release of the repair agent into the crack surfaces through capillary action, followed by the interaction between the repair agent and incorporated catalyst, thereby triggering polymerization and sealing of cracks. Figure 8b elucidates the microcapsule rupturing mechanism.

In a study conducted by Nishiwaki [72], epoxy resin was used as a repair material packed into the microcapsules with the incorporation of a urea-formaldehyde formalin shell of size 20–70 µm. The microcapsules were used together with the acrylic resin. The results indicated that self-repair was successfully achieved by the microcapsules containing sodium silicate [73]. The concrete stacking was initially performed nearly to the breaking point prior to removal of the load and subsequent curing for seven days. In comparison to the reference specimen that exhibited just 10% recovery, the specimen with 2% sodium silicate microencapsulation regained its strength to a proportion of up to 26%. It was concluded that the upper strength recovery ratio was possible by increasing the concentration of the repair material. Meanwhile, Yu et al. [74] prepared single-component microcapsules with toluene-di-isocyanate (TDI) and paraffin serving as the repair agent and shell, respectively. The findings revealed that encapsulation of TDI within the paraffin shell was successful and a better self-repair capability was demonstrated by mortars containing the microcapsules.

### 5.5. Shape Memory Materials as Self-healer

The integration of functional materials such as shape memory alloys (SMAs) or shape memory polymers, into cementitious concrete structures to promote self-repair has been advocated in a number of studies [75,76,77,78]. The underlying principle is that crack formation triggers the controlled shrinkage of these materials, producing a contraction serving to seal the cracks. The shape memory effect was discovered by Chang and Read and reported that a gold–cadmium (Au-Cd) alloy exhibits a reversible phase change [78]. Since then, several SMAs with exceptional thermo-mechanical and thermo-electrical properties have been developed [79]. For example, nitinol is a highly elastic alloy that demonstrates the shape memory effect and capable of reversing to its pre-established form upon heat exposure [80]. Its super-elasticity permits to withstand a major inelastic deformation and returns to its original form once the load is removed. Song and Mo [81] produced an intelligent reinforced concrete (IRC) by employing the SMA wires. More specifically, stranded martensite wires of the SMAs were used to accomplish post-tension effects in the IRC. The strain distribution in the concrete was obtained by tracking changes in the electrical resistance of the SMA wires. This allowed detection of the cracks forming as a result of explosions or earthquakes. The SMA wire electrical heating triggered contraction thus alleviating the cracks so that the self-repair capability was effective for managing macro-cracks. The name of the IRC derived from the fact that the concrete structure is sufficiently smart to recognize self-repair.

Sakai et al. [75] studied the super-elasticity of the SMA wires for concrete beam self-repair and acknowledged the almost complete recovery of a massive crack. Meanwhile, Jefferson et al. [77] integrated the shape memory polymers (SMPs) into cementitious materials and showed that early age shrinkage, thermal effects, and/or mechanical loading can cause the formation of cracks in the cementitious matrix. The exposure to heat can activate shrinkage of the incorporated SMA tendons, which in turn generates quantifiable compressive stresses throughout the closed crack surfaces, thus promoting crack healing. This kind of mechanism of crack closure can make concrete structural components perform better in terms of self-repair and durability. The deduction derived from the findings was that crack closure and weak pre-stressing in post-tensioned mortar beams can be successfully addressed based on incorporation of parallel polymer tendons subjected to shrinking. Numerous screening tests indicated that the tendons of highest efficiency are the polyethylene terephthalate (PET) Shrinktite, exhibiting a shrinking potential of around 34 MPa in a controlled environment with heating up to 90 °C followed by cooling down to ambient temperature. An increase of about 25% in mortar strength was estimated to be promoted by heating plus additional curing.

### 5.6. Coating

The supplementary functionalities created by the latest material technology innovations brought the concept of “intelligent material”, which refers to the ability to effectively respond to extrinsic stimuli (e.g., temperature, light, humidity). This has led to the successful development and testing of advanced construction materials such as coatings for concrete with self-repair capabilities and particular durability qualities [82,83]. For reinforced concrete, such coatings are devised to promote steel bar self-repair and minimize deterioration due to corrosion. The newest research initiative of self-repairing coatings has the potential to make a notable contribution to the efforts to combat contemporary infrastructure degradation. Unlike standard anti-corrosion coatings in which efficiency is compromised by the slightest coating damage, self-repairing coatings are capable of recovery from damage. Thus, the efficiency of self-repairing coatings remains unaffected [84]. Therefore, this capability is likely to extend the use life of steel rebar structures considerably [85]. Chen et al. [86] first explored the use of self-repairing coatings for steel rebar. Epoxy coatings that were usually applied to the rebar structures can be substituted with self-repairing coatings, especially in the northeastern regions, to protect the rebar against the high levels of corrosion.

Identification of the strategies for preventing micro-crack formation in concrete structures (e.g., roads, bridges, etc.) has been the focus of ample research, but no definitive conclusions have been reached so far [23,87,88]. The penetration of concrete by water, de-icing salt, and air is facilitated by cracks. The sub-zero temperatures cause the expansion of frozen water within the cracks, which thus become larger in size and cause the concrete to deteriorate faster upon exposure to the road salt. However, self-repairing coatings for the protection of concrete have received far less research attention than self-repairing coatings with anti-corrosion action for the protection of metal. Several studies were conducted on self-repairing coatings that trigger self-repair in response to the extrinsic crack formation or damage. Such coatings frequently incorporate micro containers that break readily in the presence of disruptive agents. The healing agents within these containers can extend the coating’s lifespan by sealing the existing cracks. Meanwhile, the containers themselves can take various forms including polyurethane microcapsules and microfilament tubes, but they rarely impact the coating mechanical properties. Various promising findings in this field warranted an additional exploration for the practical uses [33,85,89].

Over the years, diverse organic and inorganic materials have been employed as healing agents to achieve high performance self-repairing. The protection of rebar against corrosive agents (e.g., water, salts) commonly relies on epoxy coatings. Studies have also been dedicated to the creation of polymer coatings capable of activating crack repair in response to external stimuli (e.g., heat, pH alterations). The heat-responsive coatings became successful and few of them can maintain their mechanical properties even after repeated heat cycles [90,91]. It has been reported that polyelectrolyte nanocontainer coatings can respond to changes in the pH within seconds [92]. These types of coatings have great potential for further applications, especially due to their distinct mechanical properties that can completely be restored. Particular attention has been directed towards drying oils like tung oil and linseed oil owing to their excellent repair capabilities and encapsulation [93]. Upon exposure to air, tung oil undergoes polymerization to a coating characterized by toughness, glossiness, and imperviousness [94,95,96]. Due to such properties, drying oils are widely incorporated into paints, varnishes, and printing inks. Samadzadeh et al. [97] accomplished the first encapsulation of tung oil. Assessment of the microcapsules’ pull-off strength revealed that the urea-formaldehyde microcapsules adhered to the epoxy matrix more effectively than the industrial standards. In addition, evaluation of the lifespan by immersing the damaged specimens into sodium chloride solutions yielded a positive outcome. Compared to epoxy coatings, a nine-fold extension of the use life was achieved by tung oil microcapsules following the damage.

### 5.7. Engineered Cementitious Composite

The ultra-ductile fiber-reinforced cementitious composite also called the engineered cementitious composite (ECC) is a special type of concrete that was introduced at the beginning of the 1990s. The ECC was continually refined over past two decades [98]. It is highly ductile (3%–7%) and displays a tight crack size and a relatively reduced amount of fibers that does not exceed 2% by volume [99]. The distinguishing mechanical quality of the ECC is the metal-like feature. Furthermore, the ECC can withstand heavy loading following crack formation in the context of auxiliary distortions. The self-healing notion of the dry related to the bleeding was investigated by Li et al. [100] regarding the release of chemicals capable of sealing tensile cracks with ulterior air curing. In this manner, composites without cracks can recover their mechanical properties. However, the self-repair process was admitted to lack of efficiency in case of the standard concrete, cement or fiber-reinforced concrete because the tensile crack size is challenging to control in such materials. A decrease in tensile load can promote the relentless multiplication of local breaks within the crack width, leading to rapid depletion of the repair agents. Hence, it is necessary to reduce the tensile crack width to within ten of microns for achieving a successful self-repair. The alternative is to change the mechanical properties of the composites with the use of glass pipes of extremely large size. This was highlighted by several other studies that drew attention to the significance of the crack width [101,102,103]. 

The notion of self-repair is feasible for the ECC due to the major property of tight crack size control of this material, as explored by two distinct empirical studies [30]. The first study involved the use of SEM for in situ testing of the ECC with one empty glass fiber without the healing chemicals under the condition of the applied load, while the second study involved measurement of the flexural strength of the ECC incorporated glass fibers with ethyl cyanoacrylate as the sealing agent. To assess the efficiency of the sealing mediator for repair purposes following the occurrence of deterioration in the load cycles, both studies were conducted under the materials test system (MTS) load-frames. The SEM images showed the sensing and actuation processes, while the flexural stiffness recovery was indicative of the effect of regeneration. It was admitted that prior to implementation, the additional problems must be addressed.

The self-repair capability of cementitious materials and the use of external chemicals as repair glue in concrete were investigated by Li et al. [104,105,106]. They analyzed the concrete matrix and its interaction with the exposed surroundings. The cracks were induced into the ECC before exposing it to a range of environmental conditions including water penetration and submersion, wetting and drying cycles and chloride ions attack. The results revealed almost full recovery of the mechanical and transport properties, especially for the ECC preloaded with the tensile strain below 1%. Self-repair was promoted by the minute crack size, low ratio of water to binder, and abundant fly ash (FA) content via hydration and pozzolanic mechanisms.

Waste matter and/or by-products were employed by Zhou et al. [107] for local production of ECC materials. Both slag and limestone (LS) powders were used to design a number of mixtures, which were then subjected to analysis that involved measurement of tensile strain (2–3%) and crack stiffness. Results indicated that unlike the mixtures studied by Li et al. [105], the mixtures considered by Zhou et al. [107] had a higher concentration of blast furnace slag (BFS) and LS instead of FA as well as a higher water-binder ratio (0.45–0.60). It was inferred that a significantly lower amount of un-hydrated cementitious materials with curing longer than 28 days was used than in the study by Li et al. [101]. Furthermore, compared to the ECC materials with rich FA content and low ratio of water to binder, the ECC materials with high BFS and LS content and relatively high water to binder ratio display similar self-repair capability. This conclusion was derived from the tight crack width, with the ECC self-repair depending on the availability of un-hydrated cement and additional complementary products (e.g., BFS). In fact, self-repair was promoted by a low water-cementitious material ratio and high proportion of cementitious specimen. Moreover, crack size was highlighted as important for hydration-reliant self-repair as crack sealing could be achieved with minimal use of healing agent and crack bridging from both sides was facilitated.

Through release of the healing agents, the presence of microencapsulated modules generally promoted improvement of ECC micro-crack behavior and likelihood of crack formation. Consequently, the processes of sensing and actuation were made effective via microencapsulation. As previously mentioned, a considerable importance is attached to the ECC tight crack size as it minimizes the amount of healing agents needed to seal cracks and makes it easier for these agents to bridge cracks from both sides. In short, the self-repair capability of the ECC is greater compared to standard cements because of its higher proportion of cementitious materials and lower ratio of water to binder.

### 5.8. Nanomaterials Based Self-Healing Concrete

In the concrete industry, the use of self-repairing materials is still a relatively recent innovation. These materials refer to the materials that contain cement and are capable of autonomous recovery from deterioration caused by various factors. Simultaneously, a considerable interest is currently being raised by the possibility of producing sustainable concrete using nanomaterials. Thus, concrete can be made more durable and sustainable by integrating self-repair and nanomaterial technologies [108,109]. In the context of self-repair, nanomaterials have been employed primarily to mitigate steel bar corrosion in reinforced concrete. For instance, Koleva [108] indicated that the use of nanomaterials with customized qualities, such as core-shell polymer vesicles or micelles can help reinforced concrete to perform better. However, the use of nanomaterials in concrete with self-repair capabilities has not been extensively studied.

Qian et al. [110] investigated curing under air, carbon dioxide, and water in both wet and dry state, as well as the impact of nanoclay with water employed as hydration-related inner water furnishing agent on micro-cracks. The results revealed that the addition of nanoclay and more suitable amounts of cementitious materials into the mixtures could significantly enhance the repair capability. In addition, every air-cured mixture displayed satisfactory repair as attested by the absence of final crack formation at the new site. Meanwhile, Hua [63] discovered that the addition of super absorbent polymer capsules with water as inner pool for supplementary hydration afforded the ECC a greater self-repair capacity. Furthermore, various repair products were identified over the cracked facades, but there were no obvious repaired cracks. It was concluded that the self-repair mechanism does not significantly affect the cracks. Access to sufficient water or moisture was also stressed as being important as it not only served as the reactant for supplementary hydration, but also facilitated the transport of ions.

## 6. Self-Healing in Fiber-Reinforced Concrete

The concrete composite is made of binders, fine and coarse aggregates, and short length discontinuous fibers. These fibers can significantly enhance the impact resistance, ductility, and energy absorption of concrete in addition to the higher values of splitting tensile and flexural strengths dispersed in the concrete mixture [111,112]. Many types of short fibers including metallic, polymeric, natural and carbon are used to reinforce the concrete for achieving enhanced properties. Earlier studies [113,114] on fiber reinforced cementitious composites showed an appreciable inhibition of crack propagation, thus contributing to the easy crack healing in the concrete. It was concluded that the inclusion of fibers in the concrete matrix can enhance their engineering properties. Some studies [113,114] evaluated the feasibility of fiber-reinforced concrete as immobilizers wherein self-healing agents such as bacteria were used. Rauf et al. [114] showed that natural fibers (coir, flax, and jute) can be used to carry bacterial spores for self-healing of the concrete. For this purpose, the calcite precipitation bacteria, namely Bacillus subtilis KCTC-3135^T^, Bacillus cohnii NCCP-666, and Bacillus sphaericus NCCP-313, were included in the concrete matrix along with the calcium lactate pentahydrate and urea as the organic nutrients. It was found that natural fibers are capable of substantial immobilization of bacterial spores. In addition, the flax fibers provided a better protection to the bacteria with improved crack-healing and regained compressive strength. 

Figure 9 shows the microstructure of three selected concrete specimens incorporated with the bacteria and natural fibers (B. subtilis with jute (Figure 9a–d), B. cohnii with flax (Figure 9e–h), and B. sphaericus with coir fibers (Figure 9i–l)). As a carrier, the flax fibers provided a better protection to the bacteria, indicating an efficient crack-healing and pore-filling ability. Zhu et al. [115] developed some sustainable engineered cementitious composites (ECCs) by incorporating limestone calcined clay cement and polypropylene fibers. The results showed that the proposed concrete can attain an efficient recovery of the composite tensile ductility and ultimate tensile strength through the self-healing mechanism. It was established that the inclusion of fibers in the self-healing concrete matrix can positively affect the mechanical properties and self-healing efficiency of the studied concretes. 

## 7. Self-Healing Evaluation Methods

Diverse tests and techniques have been devised to assess the efficiency of different self-repair mechanisms. Several factors that determine the capability of crack repair include the width and age of cracks and type of curing [116,117]. In most studies, microscope-based visual observation is employed to establish the nature of the wide sealed cracks [37,118]. Other approaches for crack detection include digital imaging, extremely pixelated camera photographs, and X-ray computed tomography [119]. Meanwhile, the quality of the hardened concrete is assessed via structural tests at the macro-, micro-, and nano-level. Chloride penetration, porosity, and nano-mechanical value tests can provide an insight into the self-repair performance. Some studies examined the self-repair capability of the encapsulated sodium silicate, colloidal silica, and tetraethyl orthosilicate where evaluations were made through the sorptivity and gas permeability tests [120], revealing a decrease in the sorptivity of 18% and gas permeability of 69%. Furthermore, the repair material performance has been appraised by Granger and Loukili [121] via a stiffness test. Farhayu et al. [122] showed the doubling of flexural strength in the repaired concrete compared to the control sample. Snoeck et al. [123] revealed the deposition of a white material in the crack via thermogravimetric analysis. Additional tests were undertaken in other studies to determine porosity and pore size distribution, chloride penetration, and oxygen profile.

Self-repair is considered to restore the compressive strength in a proportion of as much as 60% and it has also been reported to improve ultrasonic pulse velocity. Consequently, taking into account the test results, the strategy of biological self-repair could be beneficial for long-term durability, prolonging the use life of concrete infrastructures. Conversely, not much attention has been paid to additional key tests, including the strength of bonding between the materials deposited in the cracks and the actual concrete, the stress–strain relationship curve, and gas permeability tests. As such, different approaches should be adopted in assessing the capability of concrete self-repair. In short, a more comprehensive understanding is required regarding the materials’ deposition capacity for binding in the cracks and their structural compatibility with the components of the concretes.

Micro-tensile, bending and bulge tests were carried out to evaluate the concretes’ mechanical performance at the micro-size or smaller. Ghidelli et al. [124] stated that the nanoindentation test can be useful to evaluate the materials’ mechanical properties such as the elastic moduli and residual stresses because they can provide high resolution in the load and displacement. Diverse methods such as the micro-pillar splitting approach and micro-cantilever bending are adopted to evaluate the fracture toughness of the materials in the micro to nano-scale [125]. These methods can potentially be used to assess the self-healing performance of cementitious materials for achieving better understanding of the mechanism of self-healing, especially for materials at the nano-scale. 

## 8. Effect of Addition Self-Healing Agents on Concrete Properties

It is established that inclusion of self-healing materials in the concrete matrix can offer several advantages/disadvantages to the mechanical properties depending on the nature of the materials and self-healing process. It was found insertion of the capsules in the concrete matrix can result in voids or holes in the concrete after releasing the contained agent. These voids in turn can negatively affect the strength performance of the concrete [15]. The use of bacteria spores as the self-healing agent in the concrete was shown to reduce the compressive strength performance with loss in strength between 8 to 10%. In addition, the loss in strength increases with the increase in bacteria dosages [15,59]. The observed strength reduction was attributed to the microstructure’s alteration induced by the reduced degree of hydration and poor distribution of the hydration products because of the inclusion of nutrients and microcapsules. Algaifi et al. [126] reported that the inclusion of microbial calcium carbonate in the self-healing concrete matrix can improve the compressive strength when compared with the normal concrete specimens. Similar results were obtained by Shaheen et al. [127] wherein the mechanical properties of the prepared concrete such as the compressive strength and splitting tensile strength were enhanced when immobilization techniques for the self-healing process was utilized.

## 9. Life Cycle Analysis of Self-Healing Concrete

Ample research has been conducted on self-repair technology in the last ten years and its potential for promoting an autonomous crack healing in concrete has been highlighted. This has led to the development of several self-repair mechanisms for cementitious materials. The evaluation of various products and services in terms of their effect on the environment from the moment of development until discontinuation is based on the life cycle assessment (LCA) methodology achieved standardization as ISO 14040-14044. In essence, the LCA is geared towards determining whether self-repairing concrete is more sustainable than standard concrete. Diminished deterioration rate, prolonged use life, decreased frequency and low cost of repair over the lifespan of a concrete infrastructure are among the main advantages of self-repairing concrete. These can make concrete more sustainable as reduced repair frequency translates to a decrease in the amount of material resources and energy used, decrease in the environmentally damaging emissions associated with the manufacture and transport of materials, and reduction in the traffic changes to transport infrastructure required by the repair or reconstruction work [128,129,130,131].

Van et al. [132] discovered that chlorides can be prevented from instantly permeating through the cracks when encapsulated polyurethane precursor was used as self-repair material. Furthermore, the levels of chloride in an area with the cracks was decreased by at least 75% in the self-repairing concrete. Compared to standard concrete that has a use life in marine environments of just seven years, concrete with self-repair capabilities is usable in such environments for about 60–94 years. Lengthening of the lifespan was the main determinant of significant environmental advantages (56%–75%) according to the computations of life cycle evaluation.

## 10. Conclusions

In recent times, the production of sustainable concretes via self-healing technology became useful in construction industries worldwide. The exponential increase in the usage of the OPC caused severe environmental damages. The immense benefits and usefulness of self-healing concrete technologies were demonstrated in terms of their sustainability, energy saving traits and environmental affability. The foremost challenges, current progress and future trends of smart technology enrooted self-healing concretes were emphasized. An all-inclusive overview of the appropriate literature on smart material-based self-healing concretes allowed us to draw the following conclusions:Self-healing concretes are characterized through several significant traits such as low pollution level, cheap, eco-friendly, and elevated durability performance in harsh environmental conditions. These properties make these concretes effective sustainable materials in construction industries.The internal encapsulation and hollow fiber-activated self-healing strategies are efficient for multiple-damages repair. However, these two strategies encounter some complexity in casting and have a negative impact on the mechanical properties of the proposed concretes.The inclusion of expansive agents and mineral admixtures in the concrete demonstrated superior efficiency in the self-healing process. However, it is not very effective in the presence of multiple damages.The design of the nanomaterial-based self-healing concretes with improved performances and endurance are useful for several applications, thanks to the advancement of nanoscience and nanotechnology.The environmental pollution can considerably be reduced by implementing the high strength and durable cementitious composites fabricated using diverse nanoparticles, carbon nanotubes and nanofibers.In the domain of building and construction, production of materials via the nanotechnology route is going to play a vital role in sustainable development in the near future.Use of smart materials in concrete is advantageous in terms of the improved engineering properties of the cementitious materials, especially for the generation of self-healing and sustainable concretes.This comprehensive review is believed to provide taxonomy to navigate and underscore the research progress toward smart materials based self-healing concrete technology.

## Figures and Tables

**Figure 1 biomimetics-05-00047-f001:**
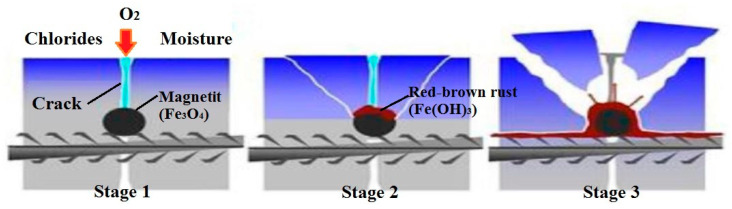
Process of aggressive solution penetration.

**Figure 2 biomimetics-05-00047-f002:**
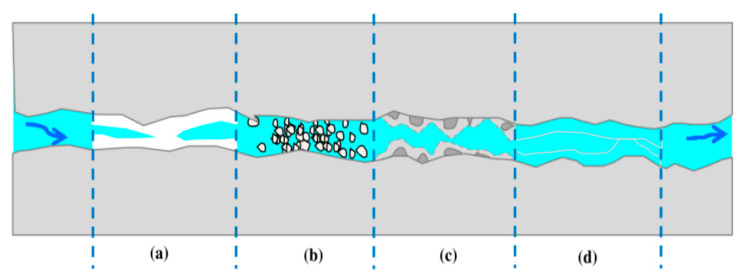
Main mechanisms of autogenous healing [37].

**Figure 3 biomimetics-05-00047-f003:**
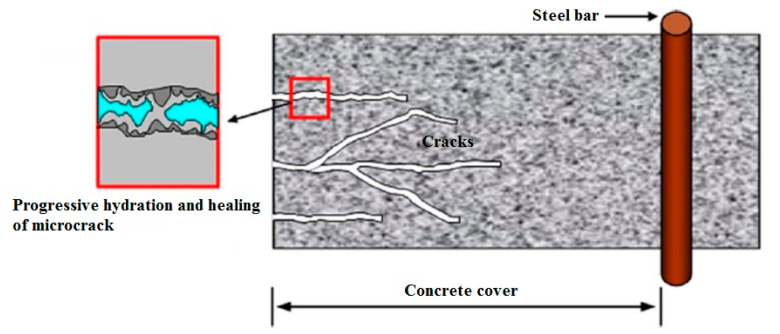
Healing of microcracks in concrete cover due to continuing hydration of unhydrated cement nuclei [43].

**Figure 4 biomimetics-05-00047-f004:**
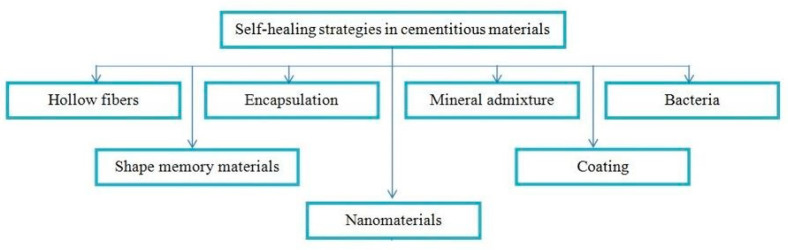
Developed strategies for self-healing in cement based materials.

**Figure 5 biomimetics-05-00047-f005:**
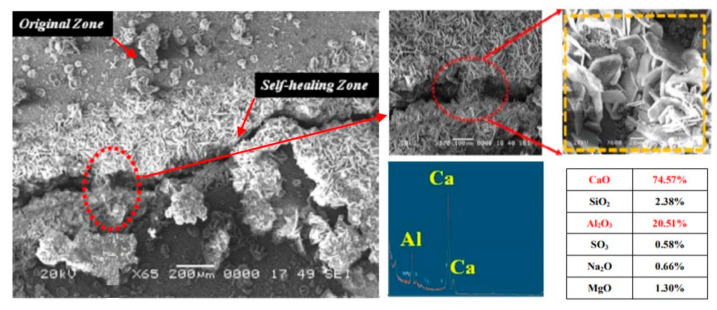
Microstructure of results between self-healing area and original area [45].

**Figure 6 biomimetics-05-00047-f006:**
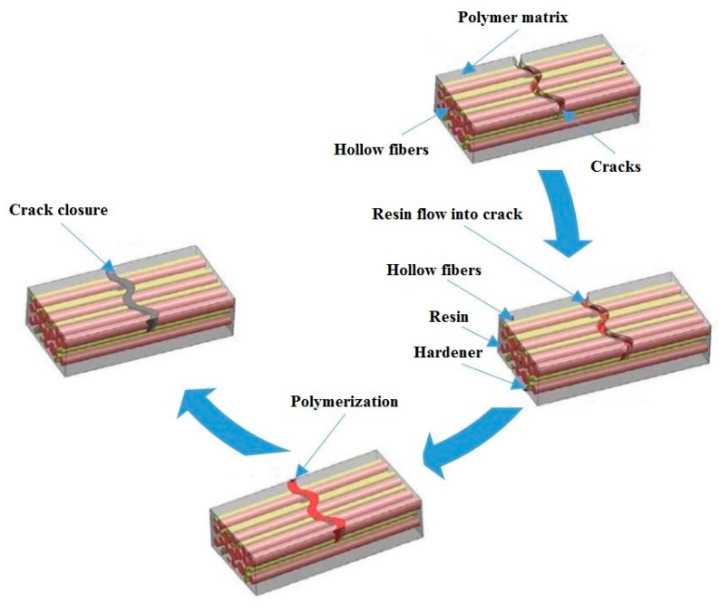
Schematic representation of self-healing processes using hollow fibers [58].

**Figure 7 biomimetics-05-00047-f007:**
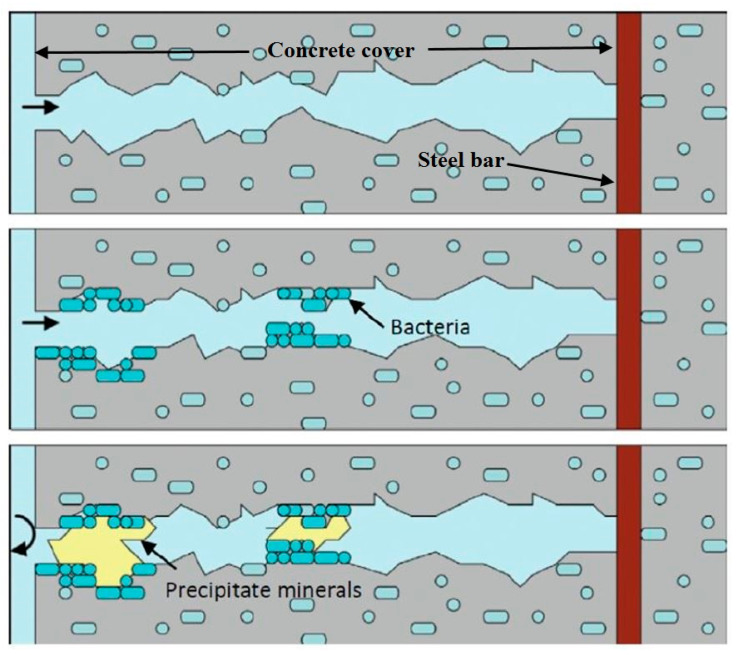
Typical crack-healing processes via immobilized bacteria in concretes [59].

**Figure 8 biomimetics-05-00047-f008:**
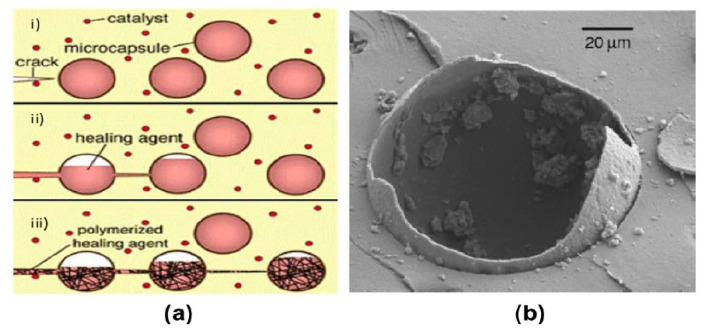
(**a**) The microencapsulation approach and (**b**) typical Field Emission Scanning Electron Microscope (FESEM) micrograph of a cracked microcapsule [69].

**Figure 9 biomimetics-05-00047-f009:**
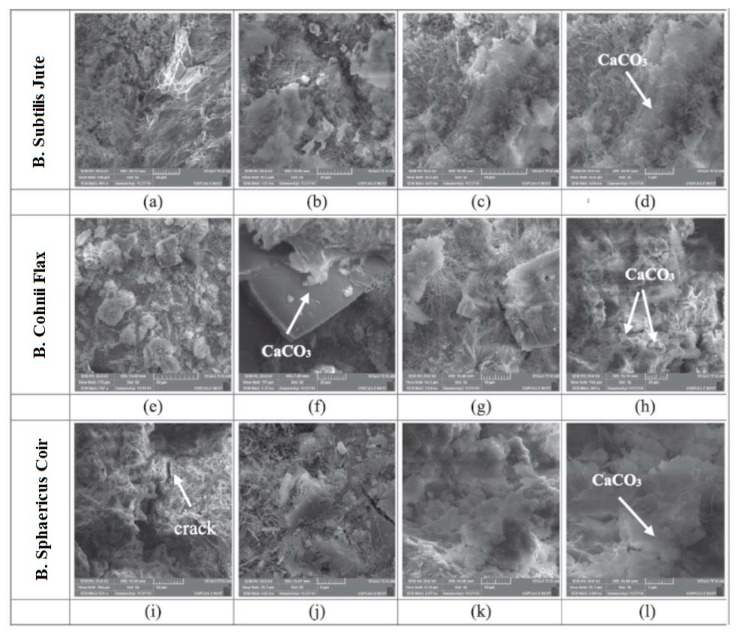
(**a**–**l**) The SEM image of the healed specimens prepared with different types of fibers [114].

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
