# Peer review of "Biomimetic Self-Healing Cementitious Construction Materials for Smart Buildings"

_biomimetics, 2020, doi:10.3390/biomimetics5040047_

Round 1
Reviewer 1 Report
The authors provide a paper dealing with the biomimetic self-healing cementitious construction materials. The paper can be of interest for Biomimetics. However, it can be significantly improved and MAJOR revisions are requested.
- The main point of the paper is that it is leaking of review an experimental characterization of the fibers embedded in the cementitious materials either coming from the literature or form authors’ work. Here, SEM images combined with post-mortem analysis or even more advanced tests (EDX, TEM, XRD) would have improved the quality of the paper, highlighting the main physical mechanisms for durability and stability of the composite.
- Connected to the previous point, the paper leaks of an analysis of the mechanical properties either of the composite (fiber+concrete) either of the single fiber. Here, there are macroscopic tests as well as one that are at the microscale. A discussion for both is required. The micro-scale techniques will enable to determine the mechanical properties on i.e. the fiber. Here, the authors can comment on the following papers doi.org/10.1016/j.matdes.2019.107762 and doi.org/10.1016/j.matdes.2016.06.003 where the authors can see how using nanoindentation techniques it will be possible to extract fracture toughness and residual stress at the nanoscale. Overall, the authors must comment in detail on the mechanical properties.
- All the part dealing with the self-healing is itself interesting. However, connected with points 1) and 2) leaks of characterization of both structure (SEM) and mechanical properties.
Reviewer 2 Report
This manuscript reviews the recent advances in self-healing cementitious construction materials for smart building. The topic is timely and important, while the current paper lacks insightful analysis of the reviewed literature. Some suggestions are below:
(1) It is a little weird to use ‘climate change’ in the title.
(2) Abstact, there is too much background information, the work and innovation of the author should be introduced in more detail.
(3) Some parts are unreadable, major technical edit of English is essential.
Errors include but not limited to the below:
Line 18, 23, 80, 125, 133, 166, 170, 366, 429, 458, 572, etc.
(4) Please specify what is the black dot in Figure 1.
(5) Sections 4 and 5 are similar and repeated, the authors may combine the two sections or restructure these parts.
(6) As a full review paper, Conclusions need to be addressed at a higher level, possibly through comparison of pros and cons of different self-healing mechanisms with more details, and discussing future challenges and trends in this field.
Round 2
Reviewer 1 Report
-
Author Response
As per reviewer' comment, we have improved the English language of manuscript.
Reviewer 2 Report
The authors have now significantly improved the manuscript while English still needs to be improved to avoid minor errors, such as Line 585, 604 etc.
Author Response
As per reviewer' comment, the have checked and modified the English language of manuscript (highlighted with Red color).